

# Association between complementary and alternative medicine use and prolonged time to conventional treatment among Thai cancer patients in a tertiary-care hospital

Adit Chotipanich[1], Chulaporn Sooksrisawat[2] and
Benjamabhon Jittiworapan[2]

[1] The Department of Otolaryngology, Chonburi Cancer Hospital, Department of Medical Services,
Ministry of Public Health, Chonburi, Thailand
[2] Nursing Unit, Chonburi Cancer Hospital, Department of Medical Services, Ministry of Public
Health, Chonburi, Thailand

Corresponding author
Adit Chotipanich, adit_c@ccc.in.th

## ABSTRACT

**Objectives:** The purpose of this study is to investigate the patterns of complementary and alternative medicine use and its association with time to conventional treatment.
**Design:** A cross-sectional study design was designed.
**Setting and participants:** The study was performed at the Chonburi Cancer Hospital, with chart reviews and interviews performed for 426 patients with various cancers between May and December 2018.
**Results:** The results indicated that 192 of the 426 patients (45.1%) reported using complementary and alternative medicines; herbal products were the most common type. Approximately 34.3% of these medicines involved unlabeled herbal products with unidentifiable components. The rates of complementary and alternative medicine use were significantly elevated for men and patients with stage IV cancer. The multivariable linear regression analysis of the relationship between factors and the time until conventional treatment was received revealed that the regression coefficient of the use of complementary and alternative medicine was 56.3 (95% confidence interval [27.9–84.6]). This coefficient reflected an additional 56.3 days of time until conventional treatment, relative to patients who did not use complementary and alternative medicine.
**Conclusions:** The present study revealed that complementary and alternative medicine use was fairly common among Thai patients with cancer and was associated with a prolonged time to receiving conventional treatment.

## INTRODUCTION

Cancer is the leading cause of death in Thailand, and is also responsible for a majority of the economic burden of disease (*Rao et al., 2010*). The cancer survival rate in Thailand is

lower than in other developed countries (*Sriamporn et al., 1995*). One factor that contributes to the relatively lower survival rate is a prolonged time until the patient undergoes cancer treatment (*Elit, 2015*), and reducing this time has become a focus of health policy-making in Thailand (*Singh, Pearlman & Kostelecky, 2017*). In 2002, Thailand implemented a universal insurance coverage scheme which has helped reduce the financial barrier to accessing treatment. However, insufficient financing remains a potential threat to healthcare service quality and sufficiency (*Paek, Meemon & Wan, 2016*). Inadequate availability of cancer treatment services is a major, but not exclusive, cause of prolonged wait times for cancer treatment, and many patient-related factors can also delay their accessing healthcare services. For example, impoverishment, distance to the treating center, lack of cancer awareness, and patient preference for complementary and alternative medicine (CAM) may preclude or delay timely access to conventional cancer treatments (*Bhosai et al., 2011*).

Although CAM is difficult to define, the World Health Organization has loosely defined CAM as "a broad set of health care practices that are not part of that country's own tradition or conventional medicine and are not fully integrated into the dominant health-care system" (*World Health Organization, 2019*). The present study focused on CAM treatments that were provided by non-medical personnel and not generally provided at Thai public hospitals.

Several demographic and health factors are associated with CAM use, such as sex, education level, income level, ethnicity, chronic diseases, and cancer (*Bishop & Lewith, 2008*). Moreover, patterns of CAM use can vary between countries, based on geographic factors, religion, government policy and regulation, and barriers to conventional treatments (*Adams et al., 2011*; *Ellison, Bradshaw & Roberts, 2011*; *Nahin, Dahlhamer & Stussman, 2010*). Previous studies have revealed that many kinds of CAM can benefit select patients (*Poonthananiwatkul et al., 2015*; *Ben-Arye et al., 2015*). For example, many herbal medicines can be used in combination with chemotherapy to prevent or ameliorate chemotherapy-related side effects, such as nausea and vomiting (*Ohnishi & Takeda, 2015*). However, improper use of CAM can also cause unexpected interactions with many prescription drugs (*Ulbricht et al., 2008*). Misleading claims and false advertising for CAM can also lead patients to select CAM instead of conventional therapy (*Kerdpon, 2001*), which can ultimately delay access to conventional therapy.

The use of CAM among patients with cancer has been studied by many researchers around the world (*Horneber et al., 2011*). However, little information is available regarding the relationship between CAM use and accessing conventional therapy among Thai patients with cancer. Thus, the present study aimed to investigate the patterns of CAM use and its association with the time until conventional cancer treatment in Thailand.

## MATERIALS AND METHODS

### Study design

This cross-sectional hospital-based study was performed at the Chonburi Cancer Hospital which is a referral center that treats cancer patients from seven provinces in eastern Thailand. Patients were non-randomly enrolled using convenience sampling during or

after conventional treatment for cancer, with data obtained using face-to-face interviews and medical record reviews between May and December 2018. The inclusion criteria were: any diagnosis of cancer (all types), received conventional cancer treatment within the last year, Thai nationality, age of ≥18 years, and sufficient cognitive function to complete the interview. The exclusion criteria were patients who received treatment for oncologic emergencies and patients who were unable to recall past therapies or related events. The study's protocol was approved by Chonburi Cancer Hospital ethics committee (IRB no. 5/2018), and all participants provided written informed consent prior to their enrollment.

## Interview and data collection

The interviews were conducted by five interviewers using a semi-structured interview protocol (Article S1). The interview protocol was reviewed by hospital specialists, and all interviewers successfully completed practice interviews with the principle researchers before interviewing the patients. The interview was described to the patients as "a survey that was designed to assess the use of alternative therapy." Patients were assured that their responses would be kept confidential and would not influence their current treatment. Patients were asked if they had ever used common CAM treatments (traditional medicine, herbal medicine, diet supplements and vitamins, acupuncture, massage, and meditation), as well as whether they had used "any other therapy not generally provided by clinics and hospitals." Patients who responded that they had used CAM treatments were also asked whether they had used any of these therapies to heal their cancer or alleviate cancer-related symptoms. Patients were categorized in the group that used CAM if the therapy was used with the intent to treat the cancer or was started after the patient became aware of the cancer diagnosis. However, herbal medicines that were approved by the Thai Food and Drug Administration and were used with a doctor's prescription were not considered CAM treatments.

Cancer stage, time until conventional treatment, and conventional treatment details were collected via the medical record reviews. The time until conventional treatment was defined as the interval between the date of diagnosis and the first date of conventional cancer treatment at the Chonburi Cancer Hospital. The date of diagnosis was defined as the day on which the diagnostic biopsy was performed. In cases with treatments performed at both the referring hospital and the Chonburi Cancer Hospital, the time until conventional treatment was calculated starting from the last day of the initial treatment at the referring hospital. Conventional treatments were categorized as surgery, chemotherapy, and radiotherapy, with concurrent chemoradiotherapy categorized as radiotherapy.

Classification of the herbal medicines and diet supplements was based on the presence or absence of registration with the Thai Food and Drug Administration. Fresh herbs, decoctions, infusions, and unauthorized commercial products were categorized as "homemade products."

## Statistical analysis

Data were analyzed using SPSS software (version 17.0; SPSS Inc., Chicago, IL, USA). Pearson's Chi-square test was used to compare categorical variables between the groups

that did and did not use CAM. Differences were considered statistically significant at *p*-values of <0.05. Associations and the influences of various factors on time to conventional treatment were analyzed using univariable and multivariable linear regression analyses. Regression coefficients reflected the additional days of time until conventional treatment, relative to the reference category for each variable.

## RESULTS

Between May and December 2018, 426 patients were enrolled, including 192 patients (45.1%) who reported using CAM. Among the 192 patients, 53 patients (27.6%) reported using CAM concurrent with or after conventional treatment. Approximately 33.6% of patients reported that they were satisfied with the CAM therapies, 61.1% reported no noticeable effect, and 5.4% reported that they were disappointed with the CAM therapies. Significant differences according to the use or non-use of CAM were observed in sex, cancer stage, and cancer type (Table 1).

Table 2 shows the details regarding the 26 kinds of CAM that were identified. Approximately 34.3% of these therapies involved unlabeled herbal products with unidentifiable components, and most products were produced as decoctions and infusions. The second most common therapy involved traditional Thai medicine ("Ya Mor Sang" or "Dr. Sang's medicine"), which is provided for free by a registered traditional Thai medicine practitioner. The provider claims that this product is composed of rice hulls and other herbal plants. Lingzhi mushroom was the third most frequently used therapy. Traditional Chinese medicine accounted for 23.1% of the CAM treatments. None of the patients used meditation or acupuncture.

The patients had a median overall time to conventional treatment of 50 days (interquartile range (IQR): 34.0–89.0 days). Patients who received radiation therapy had a noticeably longer time to conventional treatment (median: 78 days, IQR: 49.5–112.5 days) than patients who underwent surgery (median: 41 days, IQR: 29.0–62.0 days) or patients who received chemotherapy (median: 39 days, IQR: 28.0–64.0 days). Among patients who used CAM before conventional treatment, 20 patients (14.4%) deliberately chose to delay conventional treatment and instead selected CAM. These 20 patients were temporarily lost to follow-up, and had a median delay time of 382 days (IQR: 321.0–640.0 days), with a longest wait time of 2,067 days for a patient who had been selected for chemotherapy.

Table 3 shows the median times to conventional treatment and their associations with various factors. The multivariable analyses revealed that a prolonged time to conventional treatment was significantly associated with receiving radiation treatment, stage IV cancer, any use of CAM, and use of CAM before starting conventional treatment (Data S1).

## DISCUSSION

Thailand is one of a handful of middle income countries that freely provides all conventional cancer treatments to all citizens as part of a universal health coverage scheme (*Tangcharoensathien et al., 2018*). In this context, delays in receiving appropriate

**Table 1 Demographic and clinical characteristics of patients with and without the use of complementary and alternative medicine.**

| Characteristics | Use | Non-use | p-value |
|---|---|---|---|
| Sex | | | |
| Male | 115 (59.9%) | 107 (45.7%) | 0.004 |
| Female | 77 (40.1%) | 127 (54.3%) | |
| Ages in years | | | |
| <45 | 47 (24.5%) | 44 (18.8%) | 0.242 |
| 45–65 | 108 (56.3%) | 133 (56.8%) | |
| >65 | 37 (19.3%) | 57 (24.4%) | |
| Cancer stage | | | |
| I–III | 106 (55.2%) | 173 (73.9%) | <0.001 |
| IV | 86 (44.8%) | 61 (26.1%) | |
| Household income | | | |
| Low (<300 US$) | 87 (45.3%) | 126 (53.8%) | 0.080 |
| High (≥300 US$) | 105 (54.7%) | 108 (46.2%) | |
| Highest education | | | |
| Primary school | 137 (71.4%) | 174 (74.4%) | 0.487 |
| Secondary school or higher | 55 (28.6%) | 60 (25.6%) | |
| Cancer type | | | |
| Breast cancer | 14 (7.3%) | 41 (17.5%) | 0.040 |
| Gynecological cancer | 34 (17.7%) | 37 (15.8%) | |
| Head and neck cancer | 82 (42.7%) | 85 (36.3%) | |
| Colorectal cancer | 31 (16.1%) | 36 (15.4%) | |
| Others | 31 (16.1%) | 35 (15.0%) | |
| Referral province | | | |
| Nearby[a] | 138 (71.9%) | 172 (73.5%) | 0.707 |
| Distant[b] | 54 (28.1%) | 62 (26.5%) | |
| Treatment | | | |
| Surgery | 61 (31.8%) | 85 (36.3%) | 0.604 |
| Chemotherapy | 56 (29.2%) | 62 (26.5%) | |
| Radiotherapy | 75 (39.0%) | 87 (37.2%) | |

Notes:
[a] Nearby areas were defined as the Chonburi, Rayong, and Chachoengsao provinces (Article S2).
[b] Distant areas included Chantaburi, Trat, Sa Kaeo, Prachinburi, Samut Prakan, and other provinces outside the Eastern region (Article S2).

cancer treatments, regardless of their reason, can lead to poorer health outcomes and a higher economic burden (*Gildea et al., 2017*). While some CAM professions and products are regulated by governing bodies in Thailand, most CAM treatments remain informal or unregulated (*Pocaphan & Kondo, 2011*). Thus, given the availability of many unregulated CAM treatments, it is possible that CAM use might prolong the time to conventional treatment among Thai cancer patients. Very few studies have addressed this issue (*Mohd Mujar et al., 2017*; *Malik & Gopalan, 2002*; *Johnson et al., 2018*; *Puataweepong, Sutheechet & Ratanamongkol, 2012*), and the present study aimed to explore the relationship between CAM use and the time to conventional cancer treatment.

**Table 2 Complementary and alternative medicines that were reported by the 192 patients[a].**

| Medicine | Number of use (%) | Type | Major ingredients |
|---|---|---|---|
| Unlabeled fresh and processed herbal products | 74 (34.3%) | Homemade product | Unknown |
| "Ya Mor Sang" | 41 (20.0%) | Homemade product | Rice hulls and various Thai herbs |
| Lingzhi mushroom | 35 (16.2%) | All | *Ganoderma lucidum* |
| | 6 | Homemade product | |
| | 17 | Registered diet supplement | |
| | 5 | Registered herbal medicine | |
| | 8 | Undetermined product | |
| Porcupine flower | 9 (4.2%) | Homemade product | *Barleria prionitis* |
| "G-herb" | 7 (3.2%) | Registered herbal medicine | Various Thai herbs |
| Soursop tea | 7 (3.2%) | Homemade product | *Annona muricata L* |
| Dong-ChongXiaCao | 7 (3.2%) | Registered diet supplement | *Ophiocordyceps sinensis* |
| "Nan Chao Woei" | 5 (2.3%) | Homemade product | *Gymnanthemum extensum* |
| Crocodile blood capsule | 4 (1.9%) | Registered diet supplement | Freeze-dried crocodile blood |
| Sesame extracts | 4 (1.9%) | Registered diet supplement | *Sesamum indicum* |
| "Plu Kaow" | 3 (1.4%) | Registered herbal medicine | *Houttuynia cordata* |
| Mangosteen juice | 3 (1.4%) | Registered diet supplement | *Garcinia mangostana* |
| Commercial fruit and vegetable extract beverage | 2 (0.9%) | Registered diet supplement | Various fruits and vegetables |
| "Bai Ya Nang" | 2 (0.9%) | Registered diet supplement | *Tiliacora triandra* |
| *Aloe vera* juice | 1 (0.5%) | Registered diet supplement | *Aloe vera* |
| "Kao yen" | 1 (0.5%) | Homemade product | *Smilacaceae* |
| "Yong-Heng herbal solution" | 1 (0.5%) | Registered herbal medicine | Various Chinese herbs |
| "Luk Tai Bai" | 1 (0.5%) | Homemade product | *Phyllanthus niruri* |
| "Prai" | 1 (0.5%) | Homemade product | *Zingiber cassumunar* |
| "Sing Mo La" | 1 (0.5%) | Registered herbal medicine | *Cyrtosperma johnstonii* |
| Turmeric capsule | 1 (0.5%) | Registered herbal medicine | *Curcuma longa* |
| Abalone mushroom | 1 (0.5%) | Registered diet supplement | *Pleurotus ostreatus* |
| "Pien Tze Huang" | 1 (0.5%) | Registered herbal medicine | Traditional Chinese formulation |
| Korean ginseng | 1 (0.5%) | Homemade product | *Panax ginseng* |
| Soapberry tree | 1 (0.5%) | Homemade product | *Sapindus trifoliatus L.* |
| Fish oil | 1 (0.5%) | Registered diet supplement | – |
| Massage | 1 (0.5%) | – | – |

**Note:**
[a] A total of 24 patients reported using more than one type of complementary and alternative medicine.

Previous studies have revealed varying prevalence of CAM use among patients with cancer, which ranged from 24% to 84% (*Gross, Liu & Bauer-Wu, 2007*; *Bold & Leis, 2001*; *Kelly et al., 2000*; *Kappauf et al., 2000*; *Münstedt et al., 1996*), although this variation may be related to the target population and the definition of CAM that the researchers used. The present study revealed that 45.1% of the patients reported using CAM, which is moderate relative to the rates from previous studies. However, CAM use was significantly more common among men and patients with stage IV cancer.

**Table 3 The associations between times until conventional treatment and various factors.**

| Factors | N | Wait time (days) | | Univariable analysis | | | | | Multivariable analysis[a] | | | | |
|---|---|---|---|---|---|---|---|---|---|---|---|---|---|
| | | Median | IQR | Coef[b] | 95% CI | Beta | $R^2$ | p-value | Coef[b] | 95% CI | Beta | $R^2$ | p-value |
| Sex | | | | | | | | | | | | | |
| Male | 222 | 51 | 35.0–90.0 | 0 | | | | | | | | | |
| Female | 204 | 49.5 | 31.0–88.0 | −7.2 | [−36.7–22.2] | −0.023 | 0.001 | 0.630 | – | – | – | – | – |
| Cancer stage | | | | | | | | | | | | | |
| I–III | 279 | 47.5 | 32.5–79.5 | 0 | | | | | 0 | | | | |
| IV | 147 | 59 | 35.0–114.0 | 42.1 | [11.4–72.7] | 0.130 | 0.015 | 0.007 | 38.9 | [8.4–69.5] | 0.140 | 0.140 | 0.013 |
| Complementary and alternative medicine | | | | | | | | | | | | | |
| Non-use | 234 | 44 | 30.0–69.0 | 0 | | | | | 0 | | | | |
| Any use | 192 | 63 | 38.5–119.5 | 70.3 | [41.5–99.0] | 0.228 | 0.052 | <0.001 | 56.3 | [27.9–84.6] | 0.215 | 0.140 | <0.001 |
| Use before $T_x$ | 139 | 79 | 43.0–157.0 | 98.1 | [65.3–131.0] | 0.083 | 0.086 | <0.001 | 80.4 | [48.1–112.6] | 0.284 | 0.180 | <0.001 |
| Use after $T_x$ | 53 | 39 | 34.0–62.0 | −0.8 | [−17.0–15.5] | −0.005 | 0 | 0.926 | – | – | – | – | – |
| Conventional treatment modalities | | | | | | | | | | | | | |
| Surgery | 146 | 41 | 29.0–62.0 | 0 | | | | | 0 | | | | |
| Chemotherapy | 118 | 39 | 28.0–64.0 | 24.1 | [−11.3–59.6] | 0.083 | 0.007 | 0.181 | – | – | – | – | – |
| Radiotherapy | 162 | 78 | 49.5–112.5 | 59.6 | [30.8–88.4] | 0.229 | 0.053 | <0.001 | 51.3 | [23.3–79.3] | 0.197 | 0.140 | <0.001 |
| Income | | | | | | | | | | | | | |
| High (≥300 US$) | 213 | 47 | 30.0–80.5 | 0 | | | | | 0 | | | | |
| Low (<300 US$) | 213 | 54.5 | 35.5–95.0 | 17.2 | [−12.1–46.6] | 0.056 | 0.003 | 0.250 | 25.2 | [−6.7–57.2] | 0.097 | 0.140 | 0.121 |
| Education | | | | | | | | | | | | | |
| Higher education | 115 | 49 | 30.0–85.0 | 0 | | | | | 0 | | | | |
| Primary school | 311 | 50.5 | 34.0–90.0 | −2.5 | [−35.7–30.8] | −0.007 | 0 | 0.884 | 5.5 | [−30.5–41.4] | 0.018 | 0.140 | 0.765 |
| Referral areas | | | | | | | | | | | | | |
| Nearby area[c] | 310 | 48 | 33.0–88.0 | 0 | | | | | 0 | | | | |
| Distant area[d] | 116 | 53.5 | 36.0–90.0 | −0.7 | [−33.8–32.5] | −0.002 | 0 | 0.969 | −3.5 | [−34.5–27.5] | −0.012 | 0.140 | 0.825 |
| Health benefits | | | | | | | | | | | | | |
| Other[e] | 124 | 46 | 30.0–89.0 | 0 | | | | | 0 | | | | |
| UCS | 302 | 52.5 | 34.5–89.0 | −8.5 | [−40.9–23.9] | −0.025 | 0.001 | 0.607 | −10.0 | [−43.6–23.7] | −0.035 | 0.140 | 0.560 |

Notes:

IQR, interquartile range; Coef, coefficient; CI, confidence interval; Beta, standardized coefficient; $R^2$, coefficient of determination; $T_x$, the earliest modality of conventional treatment for each patient; UCS, universal coverage scheme.
[a] The analyses were adjusted for all of the other factors included in the multivariable analysis.
[b] The coefficient represents the additional days of waiting for each category relative to the reference category.
[c] Nearby areas included the Chonburi, Rayong, and Chachoengsao provinces (Article S2).
[d] Distant areas included Chantaburi, Trat, Sa Kaeo, Prachinburi, Samut Prakan, and other provinces outside the Eastern region (Article S2).
[e] Other health benefit schemes included civil-servant, social security, and self-pay schemes.

Herbal products were the most frequently used CAM, and previous studies have indicated that the classification of herbal medicines and diet supplements should be based on intended use, labeling, preparation, and dosage (Serafini et al., 2012). However, it would be difficult to distinguish between herbal medicines and diet supplements in the present study, as approximately two-thirds of the treatments involved homemade products without proper labeling or appropriate quality control. Furthermore, many treatments had been registered as diet supplements, but were used with the intent to treat the patient's cancer. Although many patients reported that use of these treatments helped alleviate their

symptoms, the use of treatments with low quality or at inappropriate dosages can cause unpredictable adverse effects and interactions with conventional treatments (*Pilkington & Boshnakova, 2012*; *Izzo & Ernst 2009*). Thus, there is a need to encourage the safe use of homemade herbal medicines among cancer patients.

The time to receiving radiotherapy was longer than the times to receiving surgery and chemotherapy. This is because our hospital is the only facility in the eastern Thailand that provides radiotherapy services, with only three teletherapy machines available to serve a population of 5,000,000 residents. Therefore, access to radiotherapy appears to remain a crucial problem in eastern Thailand.

Using CAM before conventional treatments was also significantly associated with the time to conventional treatment. Among patients who used CAM treatments, 20 patients (14.4%) had received appointments for conventional treatments but were temporarily lost to follow-up and instead chose to receive CAM treatments. The delay time in that group was remarkably longer than the times to conventional treatment in the other groups, which significantly affected the average time to conventional treatment in the present study. However, other factors may also explain why patients use CAM before conventional treatment, as many patients used CAM as an interim treatment before they could receive conventional treatment. In addition, a prolonged wait for conventional treatment could increase the likelihood that a patient would select CAM. These factors may help explain the association between CAM use and the prolonged time to conventional cancer treatment.

## LIMITATIONS

The present study has several limitations. First, the rate and pattern of CAM use at our center might be different from those in the general population, as we only evaluated patients who attended hospital appointments, which would exclude patients who select CAM treatments without seeking in-hospital treatment. It is also possible that the use of face-to-face interviews could have influenced the participants' responses (e.g., based on a desire to not reveal that they had used CAM treatments). Second, various factors could influence the categorization of CAM treatments as herbal products and diet supplements, especially as the ingredients were self-reported and their accuracy is reliant on the patient's recall. Moreover, traditional medicine practitioners tend to not disclose their formulations, which resulted in a large number of unlabeled products containing unidentifiable ingredients. Third, there remains the possibility that the study might not completely include other factors which could influence the time until treatment (e.g., religions and cultural perspectives). Finally, caution is warranted in inferring a causal relationship between CAM use and time to conventional treatment because data were obtained from a cross-sectional study.

## CONCLUSIONS

Our findings demonstrated that there is a relatively moderate prevalence of CAM use among Thai cancer patients, with most of the CAM treatments involving homemade

herbal products. The use of CAM was significantly associated with a prolonged time to conventional treatment. Nevertheless, these findings do not imply that CAM should be banned for all patients, although healthcare providers should recommend that patients aim to use CAM treatments that are considered safe and will not interfere with conventional treatments.

## ACKNOWLEDGEMENTS
We would like to thank Ms. Pimonwan Promsuvan, Ms Pannee Panjamnong, Mr Rangsan Chaikham, and the operating room staff at Chonburi cancer hospital for support in data collection. Additionally, we are grateful to Dr. Akarathan Jitnuyanont, the director of Chonburi cancer hospital, for his support and encouragement. We would like to thank Editage for English language editing.

### Funding
The authors received no funding for this work.

### Competing Interests
The authors declare that they have no competing interests.

### Author Contributions
- Adit Chotipanich conceived and designed the experiments, analyzed the data, contributed reagents/materials/analysis tools, prepared figures and/or tables, authored or reviewed drafts of the paper, approved the final draft.
- Chulaporn Sooksrisawat conceived and designed the experiments, performed the experiments, approved the final draft.
- Benjamabhon Jittiworapan conceived and designed the experiments, performed the experiments, approved the final draft.

### Ethics
The following information was supplied relating to ethical approvals (i.e., approving body and any reference numbers):
  The study's protocol was approved by Chonburi Cancer Hospital Ethics Committee (No. 5/2018).

### Data Availability
  The raw measurements are available in the Supplemental Files.

### Supplemental Information
Supplemental information for this article can be found online at http://dx.doi.org/10.7717/peerj.7159#supplemental-information.

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
