# Peer review of "Association between complementary and alternative medicine use and prolonged time to conventional treatment among Thai cancer patients in a tertiary-care hospital"

_PeerJ, doi:10.7717/peerj.7159_

## Round 0.1 · original submission · Major Revisions

· Academic Editor

Major Revisions

This is a cross-sectional survey, which examined the prevalence of a past history of unconventional treatment and its association with duration of untreated cancer in a sample of Thai patients with cancer. Our esteemed reviewers have provided helpful comments for the current paper, the authors may consider these comments and revise it accordingly. I also have a concern over the term used "unconventional therapy", but after reviewing the paper, CAM might be appropriate here, because the former term is more broad and has negative attitude. It can include worshiping gods and consulting palm readers. Although the current study focused on the possible delay in treating cancer due to unconventional therapy, unconventional therapy is not all bad for the health of patients with cancer, e.g., herbs may reduce pain associated with cancer, but can not cure cancer. The second concern is the title, which seems over-stated, because the current study did not recruit patients from all or a random sample of hospitals in eastern Thailand. The study site is a referral center hospital, but patients living at homes and receiving treatment in other hospitals were not included.

·

Basic reporting

Thank you for giving me the opportunity to review the article. The authors conducted a survey to reveal the use of alternative therapy among patients with cancer in eastern Thailand. I thought that the topic is interesting, but additional information mainly on the study methods should be provided. Comments have been listed below.

1. Basic reporting
Major Comments:
Title:
1. The word “unconventional” should be replaced with “alternative”. It because that the authors used the word “alternative” in their questionnaire.
Abstract:
2. L30: The number and % of the most common type of alternative medicine (i.e. herbal products) should be added.
3. L35-36: The most important result of the multivariable regression analysis is better to be added.
Introduction:
4. L48: References about the factor which contributes to the inferior survival rate should be added.
5. L53: A reference such as a journal article or statements which provided by the government should be added.
Results:
6. Table 1: The percentage for each group should be provided. This should not be calculated using the whole number of patients (which include both use and non-use patients). For example, a characteristic sex should be shown such like the below.
Example: For the “Use” group, male 115 (59.9%), female 77 (40.1%)

Minor Comments:
Introduction:
7. Could the authors check the words “…can benefit select”? The authors mean “…can benefit for selected patients”?
Materials and Methods:
8. L82-: Not only the original version, translated (English) version of the informed consent form should be provided (as for a supplementary material which will be published with the article).

Experimental design

Materials and Methods:
9. L80: The ethical approval number and date should be added.
10. L100: How many researchers review the medical record? If two or more researchers did review, they individually perform the review process?
11. L121: Regression coefficients also should be provided as standardized regression coefficients. In addition to this, coefficients of determination is better to be presented.
12. L121-: How to select the variables which include in the analysis? These factors may not be all the factors they collected in this study.

Validity of the findings

13. L82-: One of the most important point of the experimental design, the authors should provide the structured interview form of this study. If there is no structured form, reliability of the study results can be limited.
14. L82-Is the structured interview form of this study validated before using? If not, this should be stated in the Discussion section.
15. Table 1: Did the authors only collect the variables show in the table1? All collected variables should be shown in the Materials and Methods section to reduce the arbitrariness.

Additional comments

None

Reviewer 2 ·

Basic reporting

Reporting is clear and unambiguous.

One concern I have is to use the word complementary and alternative treatment instead of unconventional as that is not the right word to describe.

Something lacking is there is insufficient information on the gaps in literature. More citation is needed especially use of CAM around the world.

Experimental design

Fit the scope of the journal. Research question is meaningful. But the authors will need to be clear how different is the thai population compared to the rest of the world to justify the need for the study.

Methods are generally ok. Last line of statistical analysis is redundant.

Validity of the findings

The findings needed to be changed.

1. Should use word associations rather than correlations.
2. Need to adjust for the variables with p<0.1 in Table 1 as they are likely confounders.

Additional comments

Generally useful article. However, there are some concerns as above:

1. Need to justify a thailand study is important
2. what has been shown in other studies
3. statistical analysis need to be redone and preferably consult a statistician.

Reviewer 3 ·

Basic reporting

The English language should be significantly improved to ensure clarity and consistency.

Experimental design

The experimental design is fine.

Validity of the findings

The authors need to clarify what the waiting time meaning for cancer patients.

Additional comments

The quality of this manuscript needs significant improvement. The major flaw of this study is that the multivariate linear regression analysis can only indicate correlation but not causality.

The introduction part needs much more details about how previous studies have done on a similar topic.

I would reject this paper for this Journal. Detailed comments please see my attachment.

Annotated reviews are not available for download in order to protect the identity of reviewers who chose to remain anonymous.

Reviewer 4 ·

Basic reporting

This is a paper describing unconventional therapy in the treatment of cancer in eastern Thailand, and the relationship between the use of unconventional therapy and waiting time for conventional therapy. I would appreciate the authors' dedication to the study design, analysis, and the manuscript.

The English language is professional, structure and figures are understandable. And the raw data is clear.

Experimental design

Please see the comments below.

Validity of the findings

Please see the comments below.

Additional comments

Major comments:
1.STROBE Statement (strengthening the reporting of observational studies in epidemiology) is a reporting standard, it will be useful to add detail information following the STROBE checklist.

2. The target population is cancer patients in eastern Thailand in the title, but the sample is cancer patients in one hospital only although it is a referral center. The representativeness is not well enough because patients refusing transfer or seeing a doctor in other places or countries after diagnoses are not included. Besides, the survey includes patients from other places not belong to eastern Thailand according to words in Table1. It could be discussed in the limitation part if possible.

3. Line 73# “Patients were non-randomly enrolled during or after conventional treatment for cancer”, the sampling method could be described in more details of how the investigators chose who will be surveyed, and what is the specific method such as systematic sampling, stratified sampling which belong to probability sampling or convenience sampling and purposive sampling which are part of nonprobability sampling.

4Line224# It would be better to narrow the range of target population instead of “patients with cancer in eastern Thailand”, because of the representativeness mentioned above.

Minor comments:

1. keywords of “cancer wait times” is not clear enough, try to change the expression as “wait time for cancer treatment” or something else. Study type could be added in the keywords list as well.

2. The introduction part briefly shows the background of the study, but medical policies are quite different all over the world so a more detailed background of the waiting time caused by the referral system is needed in this part.

3.Table 3: the definition of “use before Tx” and “use after Tx” could be explained clearer. Do you mean before or after a whole treatment course or for example one cycle chemotherapy?

4. Line 182#-183# There is no significant difference between the high-income group and the low-income group according to the survey, so it is not suitable to give a conclusion that unconventional therapy was more commonly used in the high-income group simply based on the percentage of using unconventional therapy.

5. Line 210# Not only the prevalence but also the constitution of unconventional therapy patterns should be discussed in the limitation part due to a hospital-based population.

Thanks for the opportunity of reviewing this manuscript.

---

## Round 0.2 · Major Revisions

· Academic Editor

Major Revisions

Thanks you very much for your careful and detailed revision of the paper. Our reviewers still have some concerns over this version, I suggest you consider these comments and make necessary revisions again.

·

Basic reporting

Thank you for giving me the opportunity to review the revised version of the article. The authors improved the contents of the manuscript according to the comments provided. Additional comments (AC) have been listed below.

1. Basic reporting
Major Comments:
Title:
1. The word “unconventional” should be replaced with “alternative”. It because that the authors used the word “alternative” in their questionnaire.
Reply: Agreed, we have replaced “unconventional therapy” with “complementary and alternative medicine (CAM)”
AC: This point has been adequately corrected.
Abstract:
2. L30: The number and % of the most common type of alternative medicine (i.e. herbal products) should be added.
Reply: We have added “Approximately 34.3% of these therapies involved unlabeled herbal products with unidentifiable components” in the abstract.
AC: This point has been adequately corrected.
3. L35-36: The most important result of the multivariable regression analysis is better to be added.
Reply: We have added the regression coefficient of CAM use in the abstract.
AC: The value (regression coefficient) was added, but the meaning is still unclear in the abstract. So, the authors clearly state that the value means the additional days of waiting compared with the referenced category.
Introduction:
4. L48: References about the factor which contributes to the inferior survival rate should be added.
Reply: We have added the following reference: ‘’Elit L. Wait times from diagnosis to treatment in cancer. J Gynecol Oncol. 2015;26(4):246-8.” (reference 3)
AC: This point has been adequately corrected.
5. L53: A reference such as a journal article or statements which provided by the government should be added.
Reply: We have added the following reference: “Singh T, Pearlman PC, Kostelecky B. Supporting Evidence-Based National Cancer Control Planning: The Asia-Pacific Phase II Leadership Forum. J Cancer Policy. 2017;12:75-78.” (reference 4)
AC: This point has been adequately corrected.
Results:
6. Table 1: The percentage for each group should be provided. This should not be calculated using the whole number of patients (which include both use and non-use patients). For example, a characteristic sex should be shown such like the below.
Example: For the “Use” group, male 115 (59.9%), female 77 (40.1%)
Reply: Agreed, the table 1 has been revised accordingly.
AC: This point has been adequately corrected.

Minor Comments:
Introduction:
7. Could the authors check the words “…can benefit select”? The authors mean “…can benefit for selected patients”?
Reply: We have reviewed this with a professional language editor. “Select patients” indicates a treatment or intervention may benefit some patients, but not necessarily all patients.
AC: I understood the explanation provided by the authors.
Materials and Methods:
8. L82-: Not only the original version, translated (English) version of the informed consent form should be provided (as for a supplementary material which will be published with the article).
Reply: We have provided the English-translated version of the informed consent in the supplementary content.
AC: This point has been adequately revised.

Experimental design

2. Experimental design
Materials and Methods:
9. L80: The ethical approval number and date should be added.
Reply: We have added the IRB approval no. in the material and method section.
AC: This point has been adequately revised.
10. L100: How many researchers review the medical record? If two or more researchers did review, they individually perform the review process?
Reply: One investigator was responsible for medical record review. We have provided the medical record data collection form in the supplementary content.
AC: I understood the situation, thanks.
11. L121: Regression coefficients also should be provided as standardized regression coefficients. In addition to this, coefficients of determination is better to be presented.
Reply: We have added regression coefficients and coefficients of determination in the table 3.
AC: This point has been adequately revised.
12. L121-: How to select the variables which include in the analysis? These factors may not be all the factors they collected in this study.
Reply: All the factors that we had collected were shown in the manuscript. These factors were chosen because they have been found to be related to prolonged wait times in other studies.
AC: I understood the situation, thanks.

Validity of the findings

3. Validity of the findings
13. L82-: One of the most important point of the experimental design, the authors should provide the structured interview form of this study. If there is no structured form, reliability of the study results can be limited.
Reply: We used semi-structured interview method in this study. The English-translated version of interview protocol has been provided in the supplementary content. We have added the following sentence in the material and method section: “The interviews were conducted by five interviewers using a semi-structured interview protocol (Article S1).”
AC: This point has been adequately revised.
14. L82-Is the structured interview form of this study validated before using? If not, this should be stated in the Discussion section.
Reply: The interview protocol was reviewed by hospital specialists, and all interviewers successfully completed practice interviews with the principle researchers before interviewing the patients. We have added this information in the manuscript.
AC: I understood the situation, thanks.
15. Table 1: Did the authors only collect the variables show in the table1? All collected variables should be shown in the Materials and Methods section to reduce the arbitrariness.
Reply: Yes, all of variables were shown in the manuscript.
AC: I understood the situation, but the authors should discuss about the limited number of variables corrected in this study in the Limitation section.

Additional comments

None

Reviewer 2 ·

Basic reporting

no comment

Experimental design

no comment

Validity of the findings

no comment

Additional comments

The authors provided a satisfactory response.

Reviewer 3 ·

Basic reporting

See my detailed comments on the manuscript.

Experimental design

See my detailed comments on the manuscript.

Validity of the findings

See my detailed comments on the manuscript.

Additional comments

See my detailed comments on the manuscript.

Annotated reviews are not available for download in order to protect the identity of reviewers who chose to remain anonymous.

Reviewer 4 ·

Basic reporting

The language has been improved and and the added literature references are sufficient for readers to understand.

Experimental design

In the translated information sheet, the purpose of the study did not mention "association between CAM use and its association with time to conventional treatment". Whether participants fully understand with the aim of the study?

The procedure of the study is well explained and ethical compliance.

Validity of the findings

The collection of data is acceptable. Conclusions are revised and well stated.

Additional comments

This is a meaningful study and the revised version is adequate to publish.

---

## Round 0.3 · Minor Revisions

· Academic Editor

Minor Revisions

Thank you very much for your careful revision on the paper. Based on the reviewer comments, some remaining minor issues are needed to be addressed. Before the submission of the revised paper, please carefully edit the language again. Please ensure that the format of the paper has been written according to the guideline for authors of PeerJ,

·

Basic reporting

The manuscript has been revised appropriately according to the comments.

Experimental design

The manuscript has been revised appropriately according to the comments.

Validity of the findings

The manuscript has been revised appropriately according to the comments.

Additional comments

Thank you for giving me the opportunity to review the revised version of the article. The authors appropriately revised the manuscript according to the comments. I thought the manuscript can be accepted for publication in its current form.

Reviewer 3 ·

Basic reporting

.

Experimental design

.

Validity of the findings

.

Additional comments

Attached

Annotated reviews are not available for download in order to protect the identity of reviewers who chose to remain anonymous.

---

## Round 0.4 · accepted · Accept

· Academic Editor

Accept

Thank you for revisions. I think the paper can be accepted now.